# A Recipe for Improved Certifiable Robustness

**Kai Hu, Klas Leino, Zifan Wang, Matt Fredrikson**
Carnegie Mellon University
Pittsburgh, PA 15213, USA
{kaihu, kleino, zifanw, mfredrik}@cs.cmu.edu

## Abstract

Recent studies have highlighted the potential of Lipschitz-based methods for training certifiably robust neural networks against adversarial attacks. A key challenge, supported both theoretically and empirically, is that robustness demands greater network capacity and more data than standard training. However, effectively adding capacity under stringent Lipschitz constraints has proven more difficult than it may seem, evident by the fact that state-of-the-art approach tend more towards *underfitting* than overfitting. Moreover, we posit that a lack of careful exploration of the design space for Lipschitz-based approaches has left potential performance gains on the table. In this work, we provide a more comprehensive evaluation to better uncover the potential of Lipschitz-based certification methods. Using a combination of novel techniques, design optimizations, and synthesis of prior work, we are able to significantly improve the state-of-the-art VRA for deterministic certification on a variety of benchmark datasets, and over a range of perturbation sizes. Of particular note, we discover that the addition of large "Cholesky-orthogonalized residual dense" layers to the end of existing state-of-the-art Lipschitz-controlled ResNet architectures is especially effective for increasing network capacity and performance. Combined with filtered generative data augmentation, our final results further the state of the art deterministic VRA by up to 8.5 percentage points.

## 1 Introduction

Intentionally crafted perturbations (adversarial examples) have the potential to alter the predictions made by neural networks (Szegedy et al., 2014). Many methods have been proposed to improve the *robustness* of deep networks, either empirically or provably. In safety-critical domains especially, guarantees against adversarial examples are indispensable. Commonly, provable defenses provide certificates of *local robustness* to accompany a model's prediction; i.e., predictions should be guaranteed to be consistent within an $\ell_p$-norm-bounded $\epsilon$-ball around the input. The success of robustness certification techniques is measured by the verified robust accuracy (VRA)—the fraction of points with correct predictions that are proven to be $\epsilon$-locally robust.

To date, a look at the public robustness certification leaderboard (accessed Sept. 2023) shows that the best results are achieved by variants of *Randomized Smoothing* (RS) (Carlini et al., 2022; Cohen et al., 2019b; Jeong et al., 2021; Salman et al., 2019; Yang et al., 2021). However, there are two primary limitations associated with RS. To begin with, RS only offers a *probabilistic* guarantee, typically configured to have a 0.1% false positive certification rate. Perhaps more importantly, the inference of RS involves substantial computational overhead—this limitation is significant enough that these methods are typically tested on only a 1% subset of the ImageNet validation dataset due to timing constraints.

Another successful family of methods perform certification using Lipschitz bounds (Araujo et al., 2023; Hu et al., 2023; Leino et al., 2021; Trockman & Kolter, 2021; Wang & Manchester, 2023). Essentially, the Lipschitz constant of the neural network provides a bound on the maximum change in output for a given input perturbation, making it possible to certify local robustness. Compared with RS-based methods, Lipschitz-based methods can provide *deterministic* certification, and are efficient enough to perform robustness certification at scale, e.g., on the full ImageNet (Hu et al., 2023). While Lipschitz-based methods are promising in terms of both deterministic certification and efficiency, there is a noticeable performance gap between these methods and RS-based methods. It is *not* established, however, that this discrepancy is tied to a fundamental limitation of deterministic certification. In this work, we aim to narrow the gap between Lipschitz-based and RS-based methods.

One important avenue for improving the performance of Lipschitz-based certification is through increasing model capacity (ability to fit data). Bubeck & Sellke (2021) have shown that robust classification requires more capacity than is necessary for standard learning objectives, and Leino (2023) has shown more specifically that further capacity is required for tight Lipschitz-based certification. But while increasing model capacity for standard training is trivial—adding more blocks/layers, increasing the network width and using self-attention mechanisms are all possible approaches—in Lipschitz-based certified training, the picture is more nuanced because the network's Lipschitz constant is tightly controlled, limiting the function's expressiveness. Thus, even models with many parameters may still *underfit* the training objective.

In addition, we find that an apparent limitation preventing prior work from discovering the full potential of Lipschitz-based certification stems from the framing and evaluation setup. Specifically, most prior work is framed around a particular novel technique intended to supersede the state-of-the-art, necessitating evaluations centered on standardized benchmark hyperparameter design spaces, rather than exploring more general methods for improving performance (e.g., architecture choice, data pipeline, etc.). Although we introduce several of our own innovations, we present this work as more of a "master class" on optimizing Lipschitz-based robustness certification that draws from and synthesizes many techniques from prior work to achieve the best overall performance. This angle lets us explore design choices meant to be synergistic to the overall Lipschitz-based approach, rather than restricting us to choices tailored for head-to-head comparisons.

This work provides a more comprehensive evaluation to illuminate the potential of Lipschitz-based certification methods. First and foremost, we find that by delving more thoroughly into the design space of Lipschitz-based approaches, we can improve the state-of-the-art VRA for deterministic certification *significantly* on a variety of benchmark datasets, and over a range of perturbation sizes. In the process, we propose a number of additional techniques not already used by the prior literature that contribute to these large performance improvements. That is, our results are achieved using a combination of design optimization, novel techniques, and synthesis of prior work.

After covering the relevant background in Section 2, we begin in Section 3 with a brief survey of the design space for Lipschitz-based certified training, focusing on three key components: (1) architecture choice, (2) methods for controlling the Lipschitz constant, and (3) data augmentation. First, we cover the various architecture innovations and building blocks that have been used in the prior literature. Based on an analysis of the challenges faced by existing work, and motivated by the goal of efficiently increasing network capacity, we propose additional directions to explore along the architecture axis, including two novel network building blocks. Next, we provide an overview of the existing methods used for controlling the Lipschitz constant during training, and propose one of our own that can be combined with other approaches. Third we discuss the role data augmentation plays in training high-capacity models. Specifically, we cover DDPM (Karras et al., 2022), which prior work has found helpful for certified training, and propose an alteration to the typical augmentation strategy that we find further boosts performance. Section 4 provides an in-depth evaluation that explores along the three dimensions identified in Section 3, shedding light on the most promising design choices, and demonstrating the significant performance improvements we achieve in this work. Finally, Section 5 concludes the paper.

## 2 BACKGROUND

The problem of providing provable guarantees against adversarial examples is typically formalized using *local robustness*, which guards against the specific class of *small-norm adversarial examples*. A classification model, $F(x) = \operatorname{argmax}_i f(x)_i$, is $\epsilon$-locally robust at point $x$ if

$$\forall x' \, . \, ||x - x'||_p \leq \epsilon \Longrightarrow F(x) = F(x').$$

Given an upper bound, $K$, of the Lipschitz constant of $f$, which is an upper bound of $||f(x)-f(x')||/||x-x'||$, we can bound the maximum impact of an $\ell_p$-norm-bounded perturbation. Namely, if $j = F(x)$ is the network's prediction at $x$, $F$ is locally robust at $x$ if $\forall i \neq j \, . \, f(x)_j - \sqrt{2}K\epsilon > f(x)_i$. Note that a tighter variant of this certification procedure has been proposed in the literature, e.g., by Leino et al. (2021). Of course, this certification procedure may be very loose, both because it may be hard to obtain a tight bound on the Lipschitz constant, and because the Lipschitz constant provides only a worst-case analysis of the model's local behavior at $x$. The key to Lipschitz-based certification is that both of these issues are mitigated by the fact that the Lipschitz constant is tightly controlled (either by hard constraints or by regularization), and the certification procedure is incorporated into the model's loss function.

Lipschitz-based certification methods in the literature have a few common elements. First, all use essentially the same aforementioned certification procedure. Second, the Lipschitz bound for the entire network is bounded using a product of the layer-wise Lipschitz constants. The key variation between methods is how they perform what we will call *Lipschitz control*, which ensures that (1) the Lipschitz bound does not explode, and (2) that the learned function can be (reasonably) tightly certified using the procedure above. We characterize these variations further in Section 3.2.

## 3 DESIGN SPACE

We now turn to brief survey and analysis of the design space for Lipschitz-based certified training. We focus on three primary axes of the design space: (1) architecture choice, (2) methods for controlling the Lipschitz constant, and (3) data augmentation, covered in sections 3.1–3. We include a discussion of what prior work has done in each axis, as well as our analysis and proposals for further exploration.

### 3.1 ARCHITECTURES

Lipschitz-based certification has typically made use of a small set of architecture building blocks that are compatible with the overall approach described in Section 2. This includes 1-Lipschitz activation functions, dense layers, convolutional layers, and residual layers (with a few variations). Modules such as pooling, batch normalization, and attention are not frequently used, either because they lead to loose Lipschitz bounds, or because they are not Lipschitz at all. While ReLU activations are 1-Lipschitz, in the context of Lipschitz-based certification, they have been ubiquitously replaced by MinMax activations, whose gradient norm preservation property has been shown to be invaluable for this setting by Anil et al. (2019).

Unfortunately, while the space of architectures may be important to explore for maximizing deterministic VRA, prior work has had relatively little exploration here, often using benchmark architectures first proposed half a decade ago. On the other hand, new methods for performing Lipschitz control that present results on larger architectures may come across as misleading, as it becomes unclear if the performance benefits come from the added capacity or the Lipschitz control method. In this work we resolve this by exploring these axes more independently.

We begin our exploration with the LiResNet architecture (Hu et al., 2023) as a reference point because it performs best on CIFAR10/100 and Tiny-ImageNet datasets. The LiResNet architecture is composed of 4 parts: (1) the stem layer, a single convolution layer, to convert images in to feature maps; (2) the backbone, a stack of several residual convolutional blocks, to extract features from the feature maps; (3) the neck, $1{\sim}2$ layers to convert the feature map into a flattened vector; and (4) the classification head for predictions. Prior to recent innovations that made residual blocks effective for Lipschitz-based certification, deep architectures were not practical. However, with the LiResNet architecture, Hu et al. were able to increase the model capacity by increasing the number of blocks $L$ and the number of channels used by the block $D$. Unfortunately, they report diminishing returns beyond a dozen or so blocks (and $D{\sim}512$), at which point the network capacity is not even enough to overfit the training dataset of CIFAR-10.

We posit that stacking the same block is less effective for adding capacity in Lipschitz-based training, where the network Lipschitz is tightly controlled. Specifically, since the Lipschitz constant is bounded by the product of all blocks' Lipschitz bound, we hypothesize that any looseness in the layer-wise bounds compounds, causing overly-deep models to become over-regularized, effectively destroying its capacity. We therefore propose exploration of additional architecture features that can more effectively add capacity beyond the baseline LiResNet architecture.

**Attention-like Mechanisms.** Attention mechanisms (Dosovitskiy et al., 2020; Vaswani et al., 2017) have shown excellent ability to improve model capabilities in standard training. However, attention cannot be directly applied to Lipschitz based training since it does not have a Lipschitz bound. One alternative is Spatial-MLP (Touvron et al., 2022; Yu et al., 2022). Convolution layers extract local features while the Spatial-MLP can extract non-local features. A combination of the two different operations may allow richer features. Let $\boldsymbol{X} \in \mathbb{R}^{C \times S \times S}$ denote the feature map with $C$ channels, and a height and width of $S$; and let $\boldsymbol{W} \in \mathbb{R}^{S^2 \times S^2}$ denote the weights. The formulation of a Spatial-MLP block is (bias ignored):

$$\boldsymbol{X}[c,h,w] = \boldsymbol{X}[c,h,w] + \sum_{p=1}^{S}\sum_{q=1}^{S}\boldsymbol{W}[hS+w,pS+q]\boldsymbol{X}[c,p,q]. \tag{1}$$

The lipschitz constant of this operation is $\|\boldsymbol{I}+\boldsymbol{W}\|_2$. We also consider using a group of Spatial-MLPs with more parameters to increase model capacity. Supposing we use $G$ groups (the number of channels $C$ should be divisible by $G$), we would have $\boldsymbol{W}_i \in \mathbb{R}^{S^2 \times S^2}, 1 \leq i \leq G$ as the weights. The formulation of a group Res-MLP block is ($k$ is the integer in $[c \cdot G/C, c \cdot G/C+1)$):

$$\boldsymbol{X}[c,h,w] = \boldsymbol{X}[c,h,w] + \sum_{p=1}^{S}\sum_{q=1}^{S} \boldsymbol{W}_k[hS+w, pS+q]\boldsymbol{X}[c,p,q]. \tag{2}$$

The lipschitz constant of this operation is $\max_i(\|\boldsymbol{I}+\boldsymbol{W}_i\|_2)$.

**Dense Layers.** Another solution is to add large fully connected (i.e., dense) layers after the neck. Early deep architectures like VGG employ this practice and recent work in Lipschitz based training also gets mileage from many large dense layers (Araujo et al., 2023).

We also propose a variation on standard dense layers inspired by the LiResNet block of Hu et al., which adds residual connections to a single convolutional by modifying the layer as $f(x) = x + \text{conv}(x)$. Analogously for a dense layer with weight matrix $W$, we can add residual connections to form what we call a *residual dense layer* as $f(x) = (W+I)x$.

### 3.2 LIPSCHITZ CONTROL

Lipschitz-based certification requires the network to have a low Lipschitz constant since an upper bound on the Lipschitz constant is used to approximate output changes from input perturbations, and if it is too large, certification becomes difficult. There are two primary categories of Lipschitz control used in the literature: (1) Lipschitz regularization and (2) Lipschitz constraints.

The prevailing Lipschitz regularization approach is GloRo training proposed by Leino et al. (2021). In this approach, the layer-wise Lipschitz constant are computed as part of the forward pass and used to incorporate Lipschitz-based certification into the training objective. Thus the gradient provides feedback to keep the Lipschitz constant under control and optimized for certification. GloRo regularization is used by Hu et al. (2023), who achieve the current state-of-the-art VRA.

A wide variety of Lipschitz constraint approaches exist, typically using special re-parameterizations that force each linear layer's weights to be orthogonal (the Lipschitz constant of an orthogonal transformation is 1). We consider several of these approaches in our design space, described below.

**Cayley transformation.** (Trockman & Kolter, 2021) For skew-symmetric matrix $V$, $W = (I+V)^{-1}(I-V)$ is orthogonal, thus $f(x;V) = Wx$ is 1-Lipschitz.

**Matrix exponential.** (Singla & Feizi, 2021) For skew-symmetric matrix $V$, $W = \exp(V)$ is orthogonal, thus $f(x;V) = Wx$ is 1-Lipschitz.

**Layer-wise Orthogonal Training Layer.** (Xu et al., 2022) For non-singular matrix $V$, $(VV^\top)^{-\frac{1}{2}}V$ is orthogonal. To obtain a differentiable inverse square root, Newton's iteration steps are performed.

**Almost Orthogonal Layer.** Prach & Lampert (2022) showed that $f(x;V) = V \text{diag}(\sum_j |V^\top V|_{ij})^{-1}x$ is 1-Lipschitz.

**SDP-based Lipschitz Layer.** Araujo et al. (2023) showed that

$$h(x;W,q) = x - 2W \text{diag}(\sum_j |W^\top W|_{ij}\frac{q_j}{q_i})^{-1}\sigma(Wx)$$

is 1-Lipschitz with 1-Lipschitz activation $\sigma(\cdot)$.

**Sandwich Layer.** Wang & Manchester (2023) showed that

$$h(x;A,B,\Phi) = \sqrt{2}A^\top \Psi \sigma(\Psi^{-1}Bx)$$

is 1-Lipschitz with 1-Lipschitz activation $\sigma(\cdot)$ if $\|2A^\top B\| \leq 1$ holds. The condition is obtained by construct a long orthogonal matrix using Cayley transformation.

In addition to the above Lipschitz-constrained layers from prior work, we propose an approach to orthogonalize weights using the Cholesky decomposition. If $\Sigma$ is a symmetric positive definite matrix, there exists a unique lower triangular matrix $L = \text{Cholesky}(\Sigma)$ such that $LL^\top = \Sigma$. Then for non-singular matrix $V$, $\texttt{SolveTriangularSystem}\big(\text{Cholesky}(VV^\top), V\big)$ is orthogonal. The motivation of this Cholesky-based orthogonalization comes from the Gram–Schmidt process for obtaining an orthogonal matrix. Namely, $\texttt{SolveTriangularSystem}\big(\text{Cholesky}(VV^\top), V\big)$ is the same as the Gram–Schmidt process result of orthogonalizing $V$. Cholesky-based orthogonalization is more numerically stable and efficient. Cholesky-based orthogonalization is typically twice as fast as Cayley transformation to obtain an orthogonal matrix. We propose the following 1-Lipschitz layer:

**Cholesky-Orthogonalized Residual Layer.** Let $V \in \mathbb{R}^{n \times n}$ be the parameter, and $W$ is the Cholesky-based orthogonalization result of $I + V$ where $I$ is the identity matrix. The layer is formulated as $f(x; V) = Wx$. Note that With the residual formula (analogous to the residual dense layer proposed in Section 3.1), the training of the model can be more effective in the case of stacking multiple such layers.

Although some studies may find certain approaches can approximate certain functions more smoothly, there is no direct theory showing one method has general advantages over others for Lipschitz control. Thus we conduct a fair comparison of all above approaches to find an optimal method empirically. We note that it is also possible to combine various methods of Lipschitz control. Although we do not try all combinations, in our experiments, we use GloRo regularization for convolutional layers while combining different Lipschitz control techniques for the dense layers. See Section 4.2 for details.

## 3.3 DATA AUGMENTATION WITH GENERATED MODELS

Prior work (Hu et al., 2023) uses IDDPM (Nichol & Dhariwal, 2021) (obtain a FID of 3.27 for CIFAR-10) to generate samples. We would like to know if the performance of certificated robustness can be improved if using generative samples of better quality. We use the elucidating diffusion model (EDM) (Karras et al., 2022) to generate new samples, which obtain a FID of 1.79 for CIFAR-10. For each dataset (CIFAR10, CIFAR100 and Tiny-ImageNet), we train the diffusion models on the corresponding training set using the settings recommended by EDM. Unless otherwise specified, the diffusion models are class-conditional thus the generated images have pseudo-labels.

We also train a standard (non-robust) classification model on each dataset. We use the ResNeXt101 (32x48d) model weakly supervised pre-trained on 940 million ($224 \times 224$) images (Mahajan et al., 2018). We freeze the backbone and only fine-tune the last classification layer with the training dataset. This model archives 94%, 86% and 82% test accuracy on CIFAR-10, CIFAR-100, and Tiny-ImageNet respectively. We use this classification model's prediction probability of the pseudo-label to score every generated image. Images with the least 20% scores are filtered.

## 4 EVALUATION

We now present our evaluation, which includes an exploration of the axes of the design space discussed in Section 3. We begin by showcasing our final result—i.e., the best configuration discovered from our design space exploration—and compare its performance to the best VRA results reported in prior work (Section 4.1). To summarize, our ultimate configuration is based on the L12W512 LiResNet architecture proposed by (Hu et al., 2023), i.e., its backbone contains 12 linear residual convolution blocks with 512 channels each. We modify this architecture by using Cholesky-based orthogonalization for the dense layers in the neck (which Hu et al. controlled using GloRo regularization on standard dense layers), and adding 8 Cholesky-orthogonalized residual dense (CHORD) layers with 2,048 neurons each to the end of the neck. We refer to this proposed architecture configuration as "CHORD LiResNet." Note that we still use GloRo regularization for Lipschitz control on the convolutional layers. This model is trained using our improved generated data augmentation pipeline (see Table 4 for details). We provide further details on the exact training parameters in Appendix B.

Next, in section 4.2, we provide an overview breaking down the various improvements we applied to reach our final configuration, followed by more detailed ablation studies comparing various Lipschitz

Table 1: This table presents the clean and verified robust accuracy (VRA) of several concurrent works and our GloRo CHORD LiResNet models on CIFAR-10/100, TinyImageNet and ImageNet datasets.

| Dataset | Method | Clean Acc. (%) | VRA (%) at $\epsilon$ | | |
|---|---|---|---|---|---|
| | | | $\frac{36}{255}$ | $\frac{72}{255}$ | $\frac{108}{255}$ |
| CIFAR-10 | GloRo (Leino et al., 2021) | 77.0 | 58.4 | - | - |
| | Local-Lip-B (+MaxMin) (Huang et al., 2021) | 77.4 | 60.7 | 39.0 | 20.4 |
| | Cayley Large (Trockman & Kolter, 2021) | 74.6 | 61.4 | 46.4 | 32.1 |
| | SOC 20 (Singla & Feizi, 2021) | 76.3 | 62.6 | 48.7 | 36.0 |
| | CPL XL (Meunier et al., 2022) | 78.5 | 64.4 | 48.0 | 33.0 |
| | AOL Large (Prach & Lampert, 2022) | 71.6 | 64.0 | 56.4 | 49.0 |
| | SLL X-Large (Araujo et al., 2023) | 73.3 | 65.8 | 58.4 | 51.3 |
| | GloRo LiResNet (+DDPM) (Hu et al., 2023) | 82.1 | 70.0 | - | - |
| | **GloRo CHORD LiResNet (+DDPM)** | **87.0** | **78.1** | **66.6** | **53.5** |
| CIFAR-100 | Cayley Large (Trockman & Kolter, 2021) | 43.3 | 29.2 | 18.8 | 11.0 |
| | SOC 20 (Singla & Feizi, 2021) | 47.8 | 34.8 | 23.7 | 15.8 |
| | CPL XL (Meunier et al., 2022) | 47.8 | 33.4 | 20.9 | 12.6 |
| | AOL Large (Prach & Lampert, 2022) | 43.7 | 33.7 | 26.3 | 20.7 |
| | SLL X-Large (Araujo et al., 2023) | 46.5 | 36.5 | 29.0 | 23.3 |
| | Sandwich (Wang & Manchester, 2023) | 46.3 | 35.3 | 26.3 | 20.3 |
| | GloRo LiResNet (+DDPM) (Hu et al., 2023) | 55.5 | 41.5 | - | - |
| | **GloRo CHORD LiResNet (+DDPM)** | **62.1** | **50.1** | **38.5** | **29.0** |
| TinyImageNet | GloRo (Leino et al., 2021) | 35.5 | 22.4 | - | - |
| | Local-Lip-B (+MaxMin) (Huang et al., 2021) | 36.9 | 23.4 | 12.7 | 6.1 |
| | SLL X-Large (Araujo et al., 2023) | 32.1 | 23.2 | 16.8 | 12.0 |
| | Sandwich (Wang & Manchester, 2023) | 33.4 | 24.7 | 18.1 | 13.4 |
| | GloRo LiResNet (+DDPM) (Hu et al., 2023) | 46.7 | 33.6 | - | - |
| | **GloRo CHORD LiResNet (+DDPM)** | **48.4** | **37.0** | **26.8** | **18.6** |
| ImageNet | GloRo LiResNet (Hu et al., 2023) | 45.6 | 35.0 | | |
| | **GloRo CHORD LiResNet (+DDPM)** | **49.0** | **38.3** | | |

control methods and data augmentation generation pipelines. Finally, in Section 4.3, we compare our method with randomized smoothing based methods, demonstrating that our work bridges the gap between deterministic and stochastic certification.

## 4.1 COMPARISON WITH PRIOR WORKS

We compare our GloRo CHORD LiResNet with the following works from the literature: Vanilla GloRo Nets with TRADES loss, Cayley, Local-Lip Net, SOC with Householder and Certification Regularization (HH+CR), CPL, SLL, Sandwich, and GloRo LiResNet, which are selected for having been shown to surpass other approaches. Table 1 presents the clean and certified accuracy with different radii of certification $^{36}/_{255}, ^{72}/_{255}$, and $^{108}/_{255}$ on CIFAR-10/100 and Tiny-ImageNet. We can see that our approach outperforms all existing architectures with significant margins on clean accuracy and certified accuracy for all values of $\epsilon$. On CIFAR-10/100, our model improves the certificated accuracy at $\epsilon = ^{36}/_{255}$ by more than 8%. We also compare the empirical robustness of the proposed method with some recent work in Section A.

To date, Hu et al. (2023) is the only work to report results on ImageNet. However they do not use generated data on ImageNet. We generated 2 million samples using guided diffusion to boost our model. Other settings are the same as those on CIFAR-10/100. With the improved model and generated data, we further improve the certification accuracy on ImageNet by 3.3%.

## 4.2 ABLATION STUDIES

**Cumulative Effects from Modifications.** Since CHORD LiResNet is based on LiResNet (Hu et al., 2023), one would be interested to see the breakdown of the improvements from each modification. Table 2 shows the results. Modifications in the grey were not applied to the final model.

Table 2: This table presents the breakdown improvement of each modification. Numbers are repored in certificated accuracy ($\epsilon = {}^{36}/_{255}$) on CIFAR-10/100 datasets.

| Modification | CIFAR-10 | CIFAR-100 |
|---|---|---|
| Baseline (LiResNet L12W512) w/o modifications | 70.0 | 41.5 |
| Adding 8 Spatial-MLP blocks after the last convolution layers | 70.2 (+0.2) | 41.2 (-0.3) |
| Inserting 1 Spatial-MLP block after every convolution layer | 70.3 (+0.3) | 41.7 (+0.2) |
| Adding 8 dense layers before classification head | 73.4 (+3.4) | 43.6 (+2.1) |
| Adding 16 dense layers before classification head | 73.7 (+3.7) | 44.0 (+2.5) |
| Using Cholesky-based orthogonalization in the neck | 74.5 (+1.1) | 44.9 (+1.3) |
| Using CHORD layers in place of the 8 dense layers | 75.6 (+1.1) | 46.5 (+1.6) |
| Adopting a better pipeline to use generated data for augmentation | 78.1 (+2.5) | 50.1 (3.5) |

Table 3: This table presents the clean accuracy and VRA using different dense layers on CIFAR-10/100 datasets. All other settings are the same.

| Dataset | Layer Choice | Clean Accuracy | VRA (%) at $\epsilon$ | | |
|---|---|---|---|---|---|
| | | | $\frac{36}{255}$ | $\frac{72}{255}$ | $\frac{108}{255}$ |
| CIFAR-10 | Regular Dense Layer (w/ GloRo regularization) | 86.1 | 77.0 | 65.6 | 52.4 |
| | Cayley Dense Layer | 86.3 | 77.2 | 65.5 | 52.1 |
| | Almost Orthogonal Layer (AOL) | 85.1 | 75.4 | 63.7 | 50.0 |
| | SDP-based Lipschitz Layer (SLL) | 85.5 | 75.6 | 65.6 | 52.3 |
| | Sandwich Layer | 85.4 | 76.1 | 64.0 | 51.0 |
| | Matrix Exponential | **87.0** | 78.1 | **66.7** | **53.6** |
| | Cholesky-orthogonalized Residual Dense (CHORD) Layer | **87.0** | **78.1** | 66.6 | 53.5 |
| CIFAR-100 | Regular Dense Layer (w/ GloRo regularization) | 60.4 | 48.2 | 36.9 | 27.0 |
| | Cayley Dense Layer | 61.3 | 49.1 | 38.0 | 28.3 |
| | SDP-based Lipschitz Layer (SLL) | 61.5 | 48.9 | 37.3 | 27.3 |
| | Sandwich Layer | 61.3 | 48.3 | 36.6 | 27.1 |
| | Matrix Exponential | 61.7 | 49.8 | **38.6** | 28.8 |
| | Cholesky-orthogonalized Residual Dense (CHORD) Layer | **62.1** | **50.1** | 38.5 | **29.0** |

We explore different ways of using spatial MLPs. As shown in Table 2, we can either add all spatial-MLP blocks after the last convolution layer or insert one spatial-MLP block after every convolution layer. However, the improvement is very limited. We also consider using spatial MLPs with groups so that it can obtain larger model capacity from more parameters. We experiment with 2,4,⋯,32 groups, and there is no great difference. Although spatial-MLP shows comparable performance to transformers and conv-net in standard training, the capacity of spatial-MLP may not be enough for certificated training in a Lipschitz bounded setting.

Adding dense layers to the model can increase the model capacity significantly. Dense layers introduce millions of parameters with fairly small computational cost. In standard training, this property can make large dense layers prone to overfitting. However, in Lipschitz-based certified training, the smoothness of the model is controlled by the network's Lipschitz constant, thus large dense layers improve the capacity (and thus VRA) without significantly overfitting. Adding more dense layers (16 layers) exhibited diminishing returns, thus we choose 8 dense layers in our final configuration.

Orthogonalizing the dense layers is another effective way to improve the model's certified robustness. However, as we mentioned earlier, we do not apply orthogonalization to convolutions since this requires applying FFTs and inverse FFTs to the feature maps, which can become expensive in large models—we discuss this further in our presentation of our ablation study on Lipschitz control mechanisms.

**An Ablation Study on the Lipschitz Control Mechanism.** We conduct a fair comparison between the various approaches to Lipschitz control discussed in Section 3.2 to find an optimal method for controlling the Lipschitz constant of the many dense layers in our modified LiResNet architecture. We use the same backbone and train the model with the same settings on the same data (original dataset plus generated data), changing only the Lipschitz control mechanism for the dense layers—for convolutions we still use Gloro-based regularization for Lipschitz control. Table 3 shows the results on CIFAR-10/100 datasets and

Table 4: This table shows the effect of different generated data augmentation pipelines. VRA of GloRo CHORD LiResNet at radius $\epsilon = {}^{36}/_{255}$ is reported.

| Sample Generator | Filtered by classification score | Real/Generated sample ratio in a batch | CIFAR-10 | CIFAR-100 |
|---|---|---|---|---|
| IDDPM | ✗ |  | 75.6 | 46.5 |
| IDDPM | ✓ | 1 : 1 | 75.9 | 47.6 |
| EDM | ✗ |  | 76.1 | 47.3 |
| EDM | ✓ |  | 76.5 | 48.5 |
| EDM | ✓ | 1 : 2 | 77.1 | 49.2 |
|  |  | 1 : 3 | 78.1 | 50.1 |
|  |  | 1 : 0 | 69.2 | 39.0 |
|  |  | 0 : 1 | 75.4 | 47.6 |

Figure 1 shows the learning curve of certificated accuracy on the test set. We find that performance is better when parameterizing the dense layer weights as orthogonal matrices (Cayley, Matrix Exp, Cholesky). Cholesky-based and Matrix Exponential orthogonalization perform similarly, but Matrix Exponential is slower. It takes 32.4, 37.8 and 51.2 seconds to train one epoch with CHORD, Cayley and Matrix Exp respectively on CIFAR-10 using the same A100 machine. We conclude that CHORD layers are the optimal Lipschitz control choice for the dense layers considering both performance and efficiency.

From Figure 1, we see that using a regular dense layer and applying GloRo-style Lipschitz regularization for the final layers performs comparatively poorly in the early training stages but surpasses SLL and Sandwich orthogonalization after ~300-600 epochs. Because the dense layers are not constrained to be orthogonal in this case, the model requires more steps to learn a nearly orthogonal transformation. Note that this is particularly pronounced in the case of large dense layers, as opposed to the convolutional layers, which also rely on GloRo regularization. Leino (2022) has shown that a reliable gradient signal for orthogonalizing a linear transformation requires more power iterations as the dimension of the eigenvector increases. For large dense layers, the eigenvectors are high-dimensional, as compared to those of convolutions, which depend only on the size of the kernel (which is typically small). Thus we expect GloRo regularization to converge more slowly on dense layers than on convolutional ones.

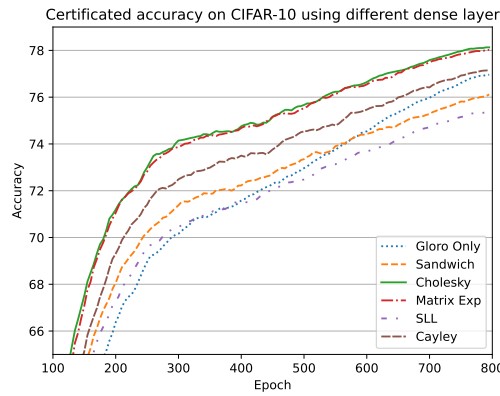

Figure 1: Certified accuracy (i.e., VRA) of our modified LiResNet architecture on CIFAR-10 using different Lipschitz control methods on the dense layers during training.

**Ablation Study on the Generated Data Augmentation.** As shown in Table 2, a better pipeline to apply generated data augmentation can improve certification robustness significantly. Table 4 shows a detailed study on the effects of different pipelines. Switching to a better generator provides consistent improvements on both datasets whether or not the sampled images are filtered. Using a stronger classification model to remove samples with least 20% low confidence pseudo-labels can also help. On CIFAR-100, the improvement is more significant. The reason is that CIFAR100 has more categories and therefore the diffusion model generates a higher proportion of images with unmatched pseudo-labels. Using these labels can harm robust training. In the second part of the table, we study the ratio of real and generated samples in a batch. We find that seeing more generated samples can significantly improve the model's certification robustness. However, if we only use generated samples to train the model (real/generated sample ratio = 0: 1), it suffers from overfitting and the performance is decreased. From this experiment, we think the reason the generated data helps is not that the generated data is of better quality, but *generated data are easier to classify on average* (training on generated data has a faster training accuracy convergence). As

Table 5: This table presents the clean and certificated robust accuracy of several *probabilistic* works and our *deterministic* GloRo CHORD LiResNet on CIFAR-10 dataset.

| method | VRA (%) measured at at $\varepsilon$ | | | |
|---|---|---|---|---|
| | $\epsilon = 0.25$ | 0.5 | 0.75 | 1.0 |
| RS (Cohen et al., 2019b) | $^{(75.0)}61.0$ | $^{(75.0)}43.0$ | $^{(65.0)}32.0$ | $^{(66.0)}22.0$ |
| SmoothAdv (Salman et al., 2019) | $^{(75.6)}67.4$ | $^{(75.6)}57.6$ | $^{(74.8)}47.8$ | $^{(57.4)}38.3$ |
| SmoothAdv (Salman et al., 2019) | $^{(84.3)}74.9$ | $^{(80.1)}63.4$ | $^{(80.1)}\mathbf{51.9}$ | $^{(62.2)}39.6$ |
| Consistency (Jeong & Shin, 2020) | $^{(77.8)}68.8$ | $^{(75.8)}58.1$ | $^{(72.9)}48.5$ | $^{(52.3)}37.8$ |
| MACER (Zhai et al., 2020) | $^{(81.0)}71.0$ | $^{(81.0)}59.0$ | $^{(66.0)}46.0$ | $^{(66.0)}38.0$ |
| DRT (Yang et al., 2021) | $^{(81.5)}70.4$ | $^{(72.6)}60.2$ | $^{(71.9)}50.5$ | $^{(56.1)}\mathbf{39.8}$ |
| SmoothMix (Jeong et al., 2021) | $^{(77.1)}67.9$ | $^{(77.1)}57.9$ | $^{(74.2)}47.7$ | $^{(61.8)}37.2$ |
| Denoised (Salman et al., 2020) | $^{(72.0)}56.0$ | $^{(62.0)}41.0$ | $^{(62.0)}28.0$ | $^{(44.0)}19.0$ |
| DDS (Carlini et al., 2022) | $^{(91.2)}\mathbf{79.3}$ | $^{(91.2)}\mathbf{65.5}$ | $^{(87.3)}48.7$ | $^{(81.5)}35.5$ |
| GloRo CHORD LiResNet (Ours) | $^{(87.0)}69.5$ | $^{(74.3)}52.2$ | $^{(70.0)}41.7$ | $^{(68.1)}35.1$ |

we mentioned before, Lipschitz-based training suffers from underfitting and much of the model capacity is used to remember hard samples, including outliers and samples very close to the decision boundary. Learning from these hard samples does not improve robustness since these samples are not naturally non-robust. When trained with generated samples (which are easier), the percent of hard samples in the dataset is decreased and the model can focus more on learning a robust decision boundary. As a contrary, generated data not always improve the performance of standard accuracy (Azizi et al., 2023) even when SOTA diffusion models are used. In the standard training setting, neural network can fit hard samples more easily. Adding too many generated samples (In Azizi et al. (2023)'s setting 6 times of the original training set), the test accuracy would decrease since most hard samples help generalization and their proportion has decreased with generated data added.

### 4.3 COMPARISON WITH RANDOMIZED SMOOTHING

To date, the methods that achieve the best certified performance are derived from randomized smoothing (RS) Cohen et al. (2019a). As we discussed, Lipschitz-based methods demonstrate advantages over RS has in terms of their efficiency and the guarantee that they provide. We provide the first comparison between these methods in Table 5. Notably, we are able to outperform several recent RS-based approaches on some or *all* certification radii.

## 5 CONCLUSION

Our primary objective in this work is to enhance the certified robustness of neural networks. We contend that a significant problem of existing Lipschitz-based models is their limited capacity, which hinders their ability to even overfit small datasets. To address this challenge, we have reexamined network architectures and basic building blocks to control network Lipschitz and have proposed three solutions to mitigate this issue. First, we showed that a combination of dense layers and convolutions can effectively expand the model's capacity. While conventional non-robust models use fewer dense layers, we find that dense layers are ideal for efficient capacity augmentation in the robust setting, which suffers less from overfitting and has a greater need for extreme capacity. Second, we introduced the Cholesky-orthogonalized Residual Desne (CHORD) Layer, which serves as an efficient building block for achieving orthogonal weights in dense layers. Although Lipschitz regularization remains best suited for Lipschitz control on convolutional layers, we find dense layers—which have very high-dimension eigenvectors—benefit from orthogonalization. Third, we explored an improved pipeline for utilizing generated data to enhance Lipschitz-based certified training. Through extensive experiments, we have demonstrated the effectiveness of our design choices. Our final results have pushed the boundaries of deterministic certified accuracy on CIFAR-10/100 datasets, surpassing the state of the art by up to 8.5 percentage points. Our method opens up a promising avenue to bridge the gap between probabilistic and deterministic certification methods, which we believe is crucial to making certification practical.

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

## A    EMPIRICAL EVALUATION OF OUR MODEL

As an evaluation of the empirical robustness of GloRo CHORD LiResNet in comparison with recent work, we use *AutoAttack* (Croce & Hein, 2020), an ensemble of diverse parameter-free attacks. Table 6 presents the results. Results of SLL and SandWich come from Wang & Manchester (2023). We find that not only does our model outperform prior work in terms of empirical robust accuracy, but additionally, when comparing Table 6 to Table 1 in Section 4.1, our model's *ceritified* robust performance (VRA) is higher than the *empirical* robust performance of the prior work under comparison.

Table 6: This table presents the clean and empirical robust accuracy of several concurrent works and our GloRo CHORD LiResNet models on CIFAR-10/100 and TinyImageNet datasets.

| Datasets | Models | Clean Accuracy | AutoAttack ($\epsilon$) | | |
|---|---|---|---|---|---|
| | | | $\frac{36}{255}$ | $\frac{72}{255}$ | $\frac{108}{255}$ |
| CIFAR-10 | CPL XL (Meunier et al., 2022) | 78.5 | 71.3 | - | - |
| | SLL X-Large(Araujo et al., 2023) | 73.3 | 70.3 | 65.4 | 60.2 |
| | GloRo CHORD LiResNet (Ours) | **87.0** | **82.3** | **76.4** | **69.6** |
| CIFAR-100 | CPL XL (Meunier et al., 2022) | 47.8 | 39.1 | - | - |
| | SLL Large (Araujo et al., 2023) | 54.8 | 44.0 | 34.9 | 27.5 |
| | Sandwich (Wang & Manchester, 2023) | 57.5 | 48.5 | 40.2 | 32.9 |
| | GloRo CHORD LiResNet (Ours) | **62.1** | **55.5** | **48.7** | **42.2** |
| TinyImageNet | SLL Medium (Araujo et al., 2023) | 30.3 | 24.6 | 19.8 | 15.7 |
| | Sandwich Medium (Wang & Manchester, 2023) | 35.5 | 29.9 | 25.3 | 21.4 |
| | GloRo CHORD LiResNet (Ours, +DDPM) | **48.4** | **42.9** | **38.1** | **33.7** |

## B    TRAINING DETAILS

We follow most of the training settings used by Hu et al. (2023) for their GloRo LiResNet model. The first difference is that we change the maximum training perturbation radius to $\epsilon_{\text{train}} = {}^{108}/_{255}$ as we need to report certificated robustness performance of this radius during inference. The original setting is $\epsilon_{\text{train}} = {}^{72}/_{255}$ as Hu et al. only report the certificated robustness performance at $\epsilon_{\text{test}} = {}^{36}/_{255}$. The second difference is the choice of generated data. We generated the generated data as described in (Hu et al., 2023). We train the model with a batch size of 1024 where 256 samples come from the original dataset and the rest 768 samples are generated. We do not change other settings including the learning rate.

## C    A NOTE ABOUT SLL RESULTS IN TABLE 3

We discovered an issue in the publicly-available SLL implementation that causes the SLL layer to be non-Lipschitz. After communicating with the authors over email, we devised an implementation that addressed the issue. Our results using SLL layers in Table 3 are based on this fix, and do not reflect the performance of the current public implementation of SLL.

