# OpenReview forum: "A Recipe for Improved Certifiable Robustness"
_ICLR.cc/2024/Conference — ICLR 2024 poster_

### Official Review · Reviewer_TVPw · 2023-10-30

**Soundness:** 3 good
**Presentation:** 3 good
**Contribution:** 3 good
**Rating:** 8
**Confidence:** 3

**Summary:**

It provides a more comprehensive evaluation to better uncover the potential of Lipschitz-based certification methods. Using a combination of novel techniques, design optimizations, and synthesis of prior work, it is able to improve the state-of-the-art VRA for deterministic certification on a variety of benchmark datasets, and over a range of perturbation sizes.

**Strengths:**

It finds that an apparent limitation preventing prior work from discovering the full potential of Lipschitz-based certification stems from the framing and evaluation setup. Specifically, most prior work is framed around a particular novel technique intended to supersede the state-of-the-art, necessitating evaluations centered on standardized benchmark hyperparameter design spaces, rather than exploring more
general methods for improving performance (e.g., architecture choice, data pipeline, etc.).


It discovers that the addition of large “Cholesky-orthogonalized residual dense” layers to the end of existing state-of-the-art Lipschitz-controlled ResNet architectures is especially effective for increasing network capacity and performance.

This work provides a more comprehensive evaluation to illuminate the potential of Lipschitz-based certification methods. It finds that by delving more thoroughly into the design space of Lipschitz-based approaches, it can improve the state-of-the-art VRA for deterministic certification significantly on a variety of benchmark datasets, and over a range of perturbation sizes. Combined with filtered generative data augmentation, the final results further the state of the art deterministic VRA by up to 8.5 percentage points.

It provides an overview of the existing methods used for controlling the Lipschitz constant during training, and propose its own method that can be combined with other approaches.

It discusses the role data augmentation plays in training high-capacity models. It covers DDPM, which prior work has found helpful for certified training, and proposes an alteration to the typical augmentation strategy that further boosts performance.

**Weaknesses:**

In section 4.3, it seems to mainly discuss the comparison with RS based methods. But Table 5 shows several other works which can achieve better performance. It is better to also discuss the comparison with these works. Currently it seems that table 5 only shows the results without detailed discussions for these works.

**Questions:**

see the weakness.

---

> ### Author Response · Authors · 2023-11-21
> **Author Response to Reviewer 9Qkw TVPw**
>
> Thanks for your thoughtful feedback. We address your concerns as follows.
>
> All methods listed in Table 5 (except our own) are based on randomized smoothing that smooths the classifier by using prediction of the input added by random noise. This property makes this line of work rely on sampling a great number of noised samples to derive probabilistic bounds of the worst prediction. Although the performance of this line of work used to be much higher than Lipschitz based methods, they have two problems: 1) very slow, not even able to certify the entire validation dataset of ImageNet, and 2) they can only provide a probabilistic guarantee. Our work is a Lipschitz based method. Thus our guarantee is deterministic and there is no overhead during inference and certification. Most importantly, our work is the first Lipschitz based method to close the performance gap with these RS based methods, showing the potential of Lipschitz based methods to give higher certification performance deterministically and efficiently.

---

### Official Review · Reviewer_9Qkw · 2023-11-01

**Soundness:** 2 fair
**Presentation:** 2 fair
**Contribution:** 2 fair
**Rating:** 6
**Confidence:** 2

**Summary:**

The work proposed some new architectures to enhance the certified robustness of neural network by reducing the Lipschitz constant. Empirical experiments showed some improvement.

**Strengths:**

1. This work studies the limitation for Lipschitz-based certification and proposed new architectures to mitigate the issue.
2. Strong empirical result: experiments showed noticeable improvement over the baseline models.

**Weaknesses:**

The authors need to include some intuitions when designing the layers.

**Questions:**

Can the authors include training time and inference time for the new architecuture?

---

> ### Author Response · Authors · 2023-11-21
> **Author Response to Reviewer 9Qkw**
>
> Thanks for your thoughtful feedback. We address the reviewer’s concerns as follows.
>
> > The authors need to include some intuitions when designing the layers.
>
> For the intuition of studying dense layers, we experimented with many networks, e.g. LiResNets and others, and found that the VRA curve reaches a plateau against the increasing network depth and width when using conventional architectures (with many conv layers). As the goal is to increase capacity, we are motivated by the fact that large dense layers easily add huge numbers of parameters. Since convolutions provide stronger regularization (e.g., weight-sharing, zeroed weights) that aligns with visual intuition, we keep them in the beginning of the model. However, as noted, Lipschitz constraint/regularization is also a strong regularizer for the model, which may rely less heavily on highly-regularized (convolutional) layers than architectures that are used in standard non-robust models. Therefore, our design focuses on dense layers.
>
> For the intuition of using Cholesky layers, we have the following findings that motivate our approach. First, we find that existing orthogonalization methods, e.g. Cayley transform or matrix exponential, have a few shortcomings. For one thing, they may suffer from the lack of expressiveness due to the strength of the constraints. Namely, they __can only represent orthogonal matrices with a positive determinant__. That is to say, they cannot represent orthogonal matrices with a determinant of -1, e.g. M=[[0, 1], [1, 0]]. However, methods like __Cholesky-based orthogonalization can represent all orthogonal matrices__, which express as much as twice of the parameter space compared to existing approaches. Additionally, orthogonalization can be somewhat slow, which limits scalability. Cholesky is also motivated by its speed.
>
> Second, regularization (GloRo) may not behave stably on large dense layers. The eigenvectors of large dense layers are very high dimensional, and must be computed accurately to propagate a good gradient signal for reducing the Lipschitz constant (and encouraging orthogonality). By contrast orthogonalization controls the Lipschitz constant directly, which makes such layers more manageable.
>
> Our use of Cholesky for orthogonalization is motivated by Gram–Schmidt process,
> which is probably the most straightforward method to orthogonalize a matrix. However, while the Gram–Schmidt process may suffer from low efficiency (need to be done recursively) and numerical instability, Cholesky-based orthogonalization is equivalent to the Gram–Schmidt process mathematically and solves the efficiency and instability issues simultaneously.
>
>
> > Can the authors include training time and inference time for the new architecture?
>
> Yes, we mentioned the training time in Sec 4.2.  Specifically, training one epoch reads 200 thousand 32-by-32 images including generated data and the inference is on 10 thousand 32-by-32 images. We use 8 A100 GPUs to train the model. The training time cost per epoch and speed using different  orthogonalization methods is listed in the following table:
>
>
>
>
>
> || training time |training speed|
> |--|--|--|
> |Cayley transform|32.3 seconds|774 images/(second * GPU)|
> |Matrix Exp|37.8 seconds|661 images/(second * GPU)|
> |Cholesky|51.2 seconds|488 images/(second * GPU)|
>
>
> The orthogonalized weights  and the Lipschitz constant of the network can be computed before inference and do not need a second computation. Thus the inference is as efficient as standard classification models. On a single A100 GPU, it takes less than 3 seconds to do inference for 10 thousand 32-by-32 images for all three methods.

---

### Official Review · Reviewer_BQAE · 2023-11-01

**Soundness:** 3 good
**Presentation:** 3 good
**Contribution:** 2 fair
**Rating:** 5
**Confidence:** 3

**Summary:**

This paper improves L2 deterministic certifiably robust training from three aspects: 1) adding additional layers; 2) using Cholesky-based orthogonal layers in the neck; 3) data augmentation with a newer diffusion model.

**Strengths:**

* By combining technical improvements on three aspects as mentioned in the summary, the paper shows a significant empirical improvement over previous works across all the datasets (e.g., +8% on CIFAR-10).
* This work provides suggestions on better settings for the robust training, in terms of model architecture with additional layers, building orthogonal layers with Cholesky decomposition, and data augmentation with a newer diffusion model.

**Weaknesses:**

* The paper looks like manually searching for settings (model architecture, orthogonal layers, diffusion model). It has engineering merits. But it does not have much novel contribution by adding more dense layers and replacing the diffusion model already used in Hu et al,, 2023 with a newer diffusion model.
* The benefits of the best choices found by the paper are not well explained. For example, the paper only explains that the Cholesky-base orthogonalization is more numerically stable and faster, but it does not explain why it can improve VRA. Therefore, the paper provides limited insights, in its current form.

**Questions:**

* How does the Cholesky-base orthogonalization help on VRA?

---

> ### Author Response · Authors · 2023-11-21
> **Author Response to Reviewer BQAE**
>
> Thanks for your thoughtful feedback! We address the reviewer’s concerns as follows.
>
>
> > But it does not have much novel contribution by adding more dense layers and replacing the diffusion model already used in Hu et al,, 2023 with a newer diffusion model.
>
> To begin with, we kindly highlight that the major contributions of this work go beyond the method we introduce, e.g. the Cholesky layers. The evaluation, results and the determination of best practices are also valuable outcomes that we would like to present to the community. We find that prior work has focused narrowly on specific techniques that are evaluated through apples-to-apples comparisons, which has left much of the potential of Lipschitz based certification on the table, as our evaluation clearly demonstrates. We now address your concerns with Hu et al. (2023).
>
> While Hu et al. (2023) primarily focused on a high-level intuition that capacity plays a crucial role in VRA, they did not delve deeply into the complexities associated with increasing model capacity. That is, without an effective way of increasing capacity, LiResNets could still suffer from diminishing returns on VRA after merely increasing the network depth and layer width. This work dives deep into the question: __how does one effectively improve capacity for certification models to improve VRA and what are the steps__? To answer that question, we investigate a range of strategies to effectively add capacity, and our findings indicate that the incorporation of additional (orthogonal) dense layers is a key factor in the improved VRA. Our work not only offers a more nuanced understanding of capacity enhancement but also paves the way for the new state-of-the-art VRAs, a large improvement compared to our baselines.
>
> Using orthogonalized dense layers turns out to be non trivial in our experiments. Although Hu et al. (2023) and other previous works in using GloRo Nets rely on penalizing the Lipschitz constants of the dense layers for robustness, our results in the Lipschitz control ablation study suggest that optimizing dense layers using GloRo loss is empirically suboptimal when the dimensions are large. By constructing orthogonalized dense layers, training GloRo Nets with much higher VRA is possible. However, the orthogonalization turns out to be not necessary for other layers, e.g. convolutions, and replacing convolutions with orthogonalized kernels is not really useful. Our novel contribution, the Cholesky-based orthogonalization for dense layers, provides an efficient way to fully exploit the performance of dense layers with large kernels and give the best performance over existing methods.
>
> Regarding the diffusion model, __optimizing the type of diffusion models only offers diminishing returns on VRA, say 0.5% ~ 0.8%__ improvement, as evidenced by Table 4. We find that __our proposed pipeline__ to use generated data augmentation, simultaneously optimizing the mix ratio between the generated images and real ones, and using a classifier to filter out some unsatisfactory generated samples, along with the diffusion model itself, results in __significant improvement up to 3.6% in Table 4__.
>
> > The paper only explains that the Cholesky-base orthogonalization is more numerically stable and faster, but it does not explain why it can improve VRA.
>
> We provide one expressiveness perspective to understand why Cholesky-based orthogonalization can improve VRA. As we mentioned, Cholesky-based orthogonalization is equivalent to the Gram–Schmidt process of a matrix. Thus __Cholesky-based orthogonalization can represent all orthogonal matrices__ (if the original matrix is the orthogonal matrix, Cholesky-based orthogonalization returns the input orthogonal matrix as the output). However, existing orthogonalization methods, e.g. Cayley transform or matrix exponential, __can only represent orthogonal matrices with a positive determinant__. That is to say,  they cannot represent orthogonal matrices with a determinant of -1, e.g. M=[[0, 1], [1, 0]].  Therefore, the parameter space using Cayley transforms or matrix exponentials is only half the parameter space using Cholesky-based orthogonalization. We will add this to the paper.
>
> As compared to regularization (GloRo), we suspect that Cholesky orthogonalization performs better on large dense layers because regularization may not behave stably on such layers. The eigenvectors of large dense layers are very high dimensional, and must be computed accurately to propagate a good gradient signal for reducing the Lipschitz constant (and encouraging orthogonality) via regularization.
>
> However, we believe that fully understanding the improvement from Cholesky-based orthogonalization requires closer looks at the loss landscape, which may be a further study. For this work, the motivation that Cholesky-based orthogonalization is more efficient than other orthogonalization methods is also important since it enables large scale training on ImageNet level datasets.

---

> > ### Comment · Reviewer_BQAE · 2023-12-03
> > **Thanks for the response**
> >
> > I think adding additional dense layers to improve the capacity is very straightforward. The authors mentioned that "Using orthogonalized dense layers turns out to be non trivial in our experiments". But there are many existing works on orthogonalized layers. By looking at the comparison with the existing methods in Table 3, the newly proposed Cholesky-based Orthogonal Layer doesn't seem to be much better than the existing methods. Therefore, I think the empirical improvement of this paper mainly comes from relatively trivial parts (especially adding additional layers), not others such as how to use orthogonalized dense layers in a new way. Thus, I'll keep my score.

---

### Meta-Review · Area_Chair_mfwZ · 2023-12-05

**Metareview:**

This paper studies the architecture of Lipschitz-controlled ResNet in the context of Lipchitz-based certification methods. Using a combinaison of tricks (novel and exported from related work) the authors were able to improve the SOTA for deterministic certification on a wide range of datasets and perturbation sizes.

The experiments are insightful and strong (in terms of SOTA). I believe this paper should be accepted.

**Justification For Why Not Higher Score:**

I am considering recommending this paper for a spotlight. More especially if I disregard the review that gave a 5 since I believe that the critism were not fair.

I would not recommend the work for an oral as I believe the topic is not the most impactful one (certified robustness is not very efficient on Imagenet)

**Justification For Why Not Lower Score:**

The paper is solid and I believe the authors addressed the concerns of Reviewer BQAE

---

### Decision · Program_Chairs · 2024-01-16

Accept (poster)